# Estimating COVID-19 incidence and prevalence using lateral flow tests in England and Scotland, 2023-2024

Martyn Fyles [1] ✉, Jonathon Mellor [1], Robert S. Paton[1], Christopher E. Overton[1,2], Alexander M. Phillips [1,3], Alex Glaser[1] & Thomas Ward [1]

SARS-CoV-2 continues to cause substantial morbidity and mortality, particularly in winter. During the SARS-CoV-2 pandemic, community prevalence surveys provided detailed monitoring of infection levels. The Winter Coronavirus (COVID-19) Infection Survey (WCIS), conducted in England and Scotland from the 14th November 2023 to the 7th March 2024, enabled the UK Health Security Agency to publish fortnightly estimates of community infection levels in England and Scotland. Unlike previous community prevalence surveys, WCIS conducted testing using Lateral Flow Device (LFD) tests, and featured a repeat testing design that enabled estimation of key epidemiological parameters. LFD tests have a substantially lower cost per unit than Polymerase Chain Reaction (PCR) tests which were used in previous SARS-CoV-2 prevalence surveys; however, they have a high false negative rate that must be accounted for to produce reliable estimates. In this manuscript, statistical methods to robustly estimate incidence and prevalence while adjusting for time-varying false negative rates are developed. This enabled timely and robust inference of the incidence and prevalence of SARS-CoV-2, stratified by age group, location and sex. Overall, the study design of WCIS overcame key limitations of earlier large-scale community prevalence studies and demonstrated the utility of LFD tests in infectious disease surveillance.

Respiratory pathogens, such as influenza and SARS-CoV-2, contribute substantially to the burden of disease in England and Scotland, particularly during the winter period[1]. Epidemic waves for respiratory pathogens can vary substantially in size from year to year[2], and consequently, reliable surveillance of respiratory pathogens during the winter period is an essential tool in public health. Surveillance of SARS-CoV-2 is particularly important, as the virus continues to evolve within a shifting epidemiological and immunological landscape[3], and its seasonal dynamics are still uncertain due to its relatively recent emergence in 2019.

In England and Scotland, the majority of respiratory pathogen surveillance is subject to a severity bias[2], e.g. reports of influenza-like illness, data from individuals who sought healthcare, and morbidity/mortality metrics. Older age groups are particularly affected by this severity bias, which can make it difficult to understand epidemic trends in the wider population. Community prevalence studies test randomly sampled individuals for infection with the pathogen of interest, which facilitates less-biased surveillance of the epidemic in the general population. In England during the SARS-CoV-2 pandemic, two large-scale community prevalence surveys, Real-time Assessment

[1]Infectious Disease Modelling Team, Modelling Division, Chief Data Officer Group, UK Health Security Agency, London, UK. [2]Department for Mathematical Sciences, University of Liverpool, Liverpool, UK. [3]Department of Electrical Engineering and Electronics, University of Liverpool, Liverpool, UK. ✉e-mail: Martyn.Fyles@ukhsa.gov.uk

of Community Transmission (REACT) and Community Infection Survey (CIS)[4–6], were conducted. These studies provided decision makers with timely, reliable, and representative surveillance of the SARS-CoV-2 epidemic, which was of vital importance during a public health emergency.

From the 14th November 2023 to the 7th March 2024, the UK Health Security Agency (UKHSA), in partnership with the Office for National Statistics (ONS), ran the Winter Coronavirus (COVID-19) Infection Study (WCIS); a community prevalence study that aimed to estimate prevalence and incidence of SARS-CoV-2 in England and Scotland. Prevalence is defined as the proportion of the population currently infected with SARS-CoV-2, and incidence is defined as the rate of new SARS-CoV-2 infections per capita and per day. During the study, UKHSA published fortnightly estimates of the incidence and prevalence in reports that were designed to be accessible to a broad audience[7], including public health officials, policy makers, and members of the public. The study design of WCIS builds upon the work of the CIS, which was performed by ONS. Previous SARS-CoV-2 community prevalence studies used Polymerase Chain Reaction (PCR) tests; however, WCIS moved to using Lateral Flow Device (LFD) tests. Additionally, WCIS introduced a repeat-testing design for participants who tested positive. This enabled the estimation of key epidemiological parameters[8] that are then used to infer the prevalence and incidence in this manuscript.

In the primary analysis of REACT and CIS, PCR test sensitivity was not estimated or accounted for. As such, these studies reported SARS-CoV-2 PCR positivity, rather than SARS-CoV-2 prevalence. Given that the PCR test is highly sensitive[9], positivity and prevalence can be similar; however, divergences may appear due to epidemic dynamics. The analysis of CIS used post-stratification to produce representative estimates, whereas REACT relied upon the ongoing recruitment of a representative sample for each study round. The model structure of REACT was therefore required to handle gaps between subsequent rounds. In our previous work[10], we combined the datasets of CIS and REACT, and further extended the methods of these studies to estimate the time-varying PCR test sensitivity, incidence rate, and infection mortality/hospitalisation risk. Inference of these additional quantities requires several epidemiological parameters, such as duration of positivity, to be estimated from repeat-testing data. REACT and CIS were not designed with the goal of collecting repeat-testing data, though CIS did collect a small amount of sparse repeat-testing data that could be used to estimate the additional parameters. However, the sparsity of the repeat-testing data necessitated the use of complex model structures during analysis. To facilitate easier and more robust estimation of key epidemiological parameters, WCIS was designed to include repeat testing for all individuals who tested positive. In our previous work[8], the WCIS repeat testing data were used to estimate key epidemiological parameters; here, we apply those parameters to infer the incidence, prevalence and time-varying false negative rate.

Previous SARS-CoV-2 community prevalence studies focused primarily on PCR tests. PCR tests are processed in a laboratory and are highly sensitive as a result, though the laboratory processing requirement makes PCR tests relatively expensive and necessitates a delay of up to several days to receive the result. Further, a community prevalence study requires sufficient laboratory capacity/infrastructure to be available throughout the study period, which is not guaranteed. In contrast, the LFD tests are self-contained units that can be performed at home, provide results in under 30 min and have a substantially lower cost per test. LFD tests, therefore, have many advantages over PCR tests in the context of community prevalence surveys. The primary disadvantage of LFD tests is a reduced test sensitivity when compared to PCR tests[11]. As LFD tests were used in this study, the lower sensitivity will cause false negative results to be observed for some tests, i.e. some individuals who are infected with SARS-CoV-2 will incorrectly receive a negative LFD test result. Therefore, it is necessary to adjust positivity for the lower sensitivity of LFD tests accurately and robustly to obtain reliable estimates of the prevalence.

Several papers have demonstrated that average cycle threshold values, a proxy for the average viral concentration in samples, vary over time due to being conditional upon the recent epidemic dynamics[12], which is known as epidemic phase bias. As LFD tests have been shown to have a reduced sensitivity to small viral concentrations[11], it is plausible that epidemic phase bias affects average LFD test sensitivity more than it affects average PCR test sensitivity. As such, to produce reliable and unbiased estimates of the prevalence, it is necessary to develop methods to account for the potentially time-varying nature of LFD test sensitivity.

This manuscript develops a Bayesian multilevel regression approach for estimating the SARS-CoV-2 incidence and prevalence over time. A convolution function approach is used, which obtains all quantities of interest as convolutions of the incidence time series. The resulting estimates of LFD test sensitivity are conditional upon the recent epidemic dynamics, and therefore, the effects of epidemic phase bias on test sensitivity are accounted for in the model. An analysis is performed to understand the potential error in the estimated prevalence had the effects of the epidemic phase bias not been adjusted for. All model components, including the epidemiological parameter estimation models that use the WCIS repeat-testing data from our previous work[8], were implemented in a single Bayesian programme allowing for full propagation of uncertainty throughout the model. Overall, the methodological improvements in the design and analysis of WCIS allowed LFD tests to be successfully utilised in community prevalence studies to produce timely epidemic surveillance with a lower cost per test.

## Results
### Prevalence
The SARS-CoV-2 prevalence in England and Scotland between the 14th November 2023 and the 7th March 2024 is provided in Fig. 1. Rapid growth in SARS-CoV-2 occurred at the beginning of December 2023 before reaching a peak prevalence of 4.54% (95% CI: 3.90–5.24) on the 22nd December 2023, after which prevalence began to rapidly decline. Between the 18th January 2024 and the 1st February 2024, the prevalence plateaued at approximately 2% before continuing to decline until the end of the study. A minimum prevalence of 0.718% (95% CI: 0.543–0.945) occurred on the 1st March 2024.

Age-stratified estimates of SARS-CoV-2 prevalence are provided in Fig. 2. The shape of the epidemic trajectory was broadly consistent across age groups, though it varied in magnitude. Prevalence was consistently highest in those aged between 35 and 44 years, with those aged 18 and 34 years a close second. For those aged 45 and above, the prevalence decreased as the group age increased. Supplementary Table 2 provides estimates of when the prevalence peaked and the corresponding peak prevalence value.

Location-stratified estimates of SARS-CoV-2 prevalence are provided in Fig. 3, with the corresponding peak prevalence dates and values provided in Supplementary Table 3. There was limited variation in the prevalence between different regions. The highest peak prevalence of 5.37% (95% CI: 4.60–6.27) occurred in London, and the North East and Scotland had the lowest peak prevalence of 3.92% (95% CI: 3.17–4.70) and 3.92% (95% CI: 3.28–4.60) respectively.

Sex-stratified prevalence is provided in Supplementary Fig. 1. Overall, there were negligible differences in prevalence between the sexes. Prevalence peaked at 4.59% (95% CrI: 3.95–5.32%) in females, and at 4.47% (95% CrI: 3.83–5.18%) in males.

Supplementary Figs. 2–4 plot the raw sample positivity against the poststratified estimates of positivity and prevalence. Differences between the sample positivity and the poststratified positivity are due to the reweighting to adjust for over-/under-representation, and the

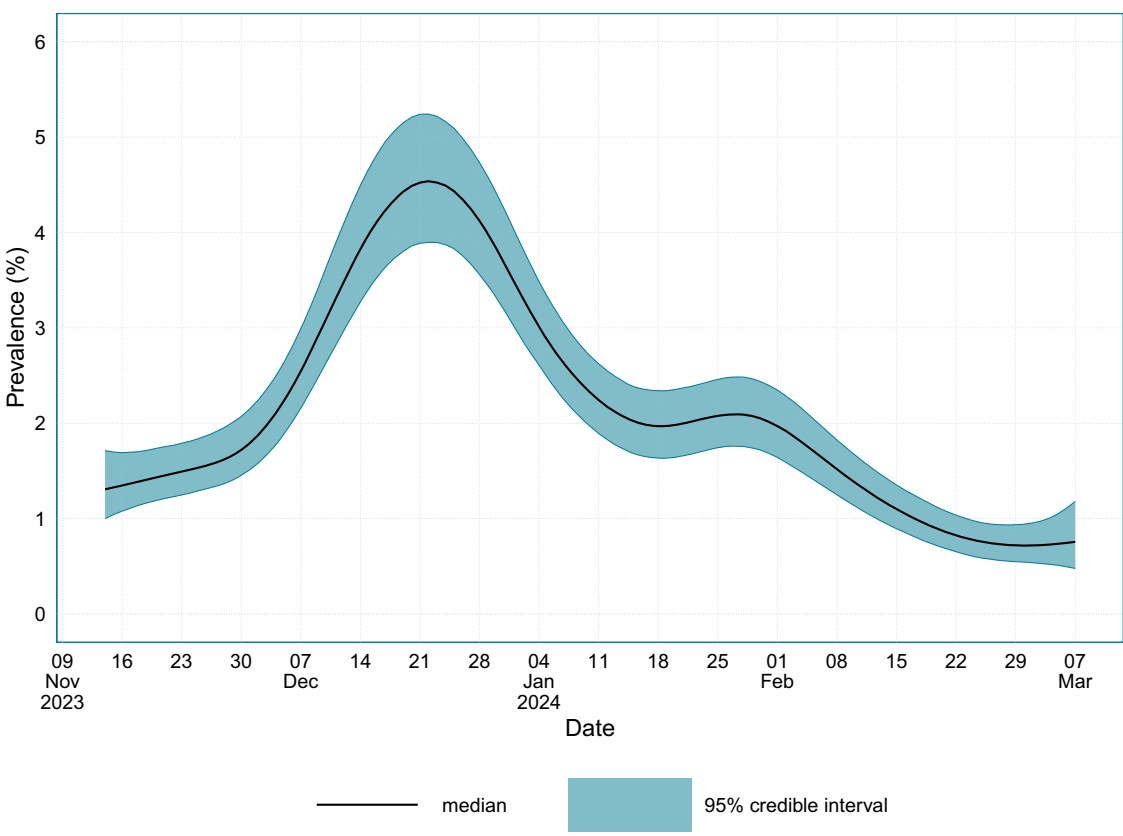

**Fig. 1 | SARS-CoV-2 prevalence in England and Scotland.** The prevalence of SARS-CoV-2 in England and Scotland between the 14th November 2023 and the 7th March 2024. Source data are provided as a Source data file.

correction for the window effects. The posterior estimates of the window effects are provided in Supplementary Fig. 5 and highlight how individuals who tested earlier in the window were more likely to test positive.

## Incidence

The estimated incidence for England and Scotland is presented in Fig. 4. In England and Scotland, incidence peaked on the 17th December 2023 at 498 (95% CrI: 429–585) new infections per 100,000 individuals per day. The results suggest that after the main epidemic peak had occurred, the incidence rate increased slightly around mid-January 2024, which would explain the plateau in prevalence that was observed during this period.

The age-stratified incidence is plotted in Fig. 5, and the corresponding timing and values of peak incidence are provided in Supplementary Table 4. In general, throughout the study period incidence was highest in the 35–44 years age group, with a peak incidence of 643 (95% CI: 540–763) new infections per 100,000 people per day, and lowest in those aged 75 years and over, with a peak incidence of 249 (95% CI: 206–299) new infections per 100,000 people per day. Around mid-January 2024, after the primary epidemic peak, an increase in the incidence rate was observed in those aged under 54 years. The location-stratified estimates of incidence are provided in Fig. 6, and the timings and values of minimum and maximum incidence are provided in Supplementary Table 5. There was limited variation in incidence across different locations. The sex-stratified incidence is plotted in Supplementary Fig. 6, again showing negligible differences between the sexes. Incidence peaked in females at 503 (95% CrI: 428–593) new infections per 100,000 individuals per day, and in males at 492 (95% CrI: 418–579).

## Average number of infections per individual

The average number of infections per individual that occurred over the whole study period is presented in Fig. 7. For England and Scotland, during the study period, there was, on average, 0.258 (95% CI: 0.224–0.298) infections per individual. The average number of infections per individual was highest in those aged 35–44 years at 0.338 (95% CI: 0.288–0.395) infections per individual, and lowest in those aged 75 years and over with 0.122 (95% CI: 0.103–0.143) infections per individual. The location with the most infections per individual was London, with 0.301 (95% CI: 0.258–0.35) infections per individual, and the location with the least infections per individual was Scotland, with 0.219 (95% CI: 0.184–0.259) infections per individual. There was little difference in the average number of infections per individual between the sexes. The time-varying average number of infections per individual over the course of the study is plotted in Supplementary Figs. 7–10.

## Sensitivity

The population average test sensitivity for England and Scotland combined is provided in Fig. 8, alongside the prevalence growth rate for England and Scotland. During the study period, the average test sensitivity in England and Scotland was 72.1% (95% CrI: 70.3, 74.0); however, due to epidemic phase bias, the average test sensitivity in England and Scotland varied substantially over the course of the study. In England and Scotland, the maximum value of the population average test sensitivity was 77.2% (95% CrI: 75.3–79.2), which occurred on the 9th December 2023, and the minimum population average test sensitivity value was 68.6% (95% CrI: 66.4–70.7), which occurred on the 4th January 2024. The population average test sensitivity was strongly correlated with epidemic growth rate, and the maximum test sensitivity value occurred when the epidemic growth rate in England and

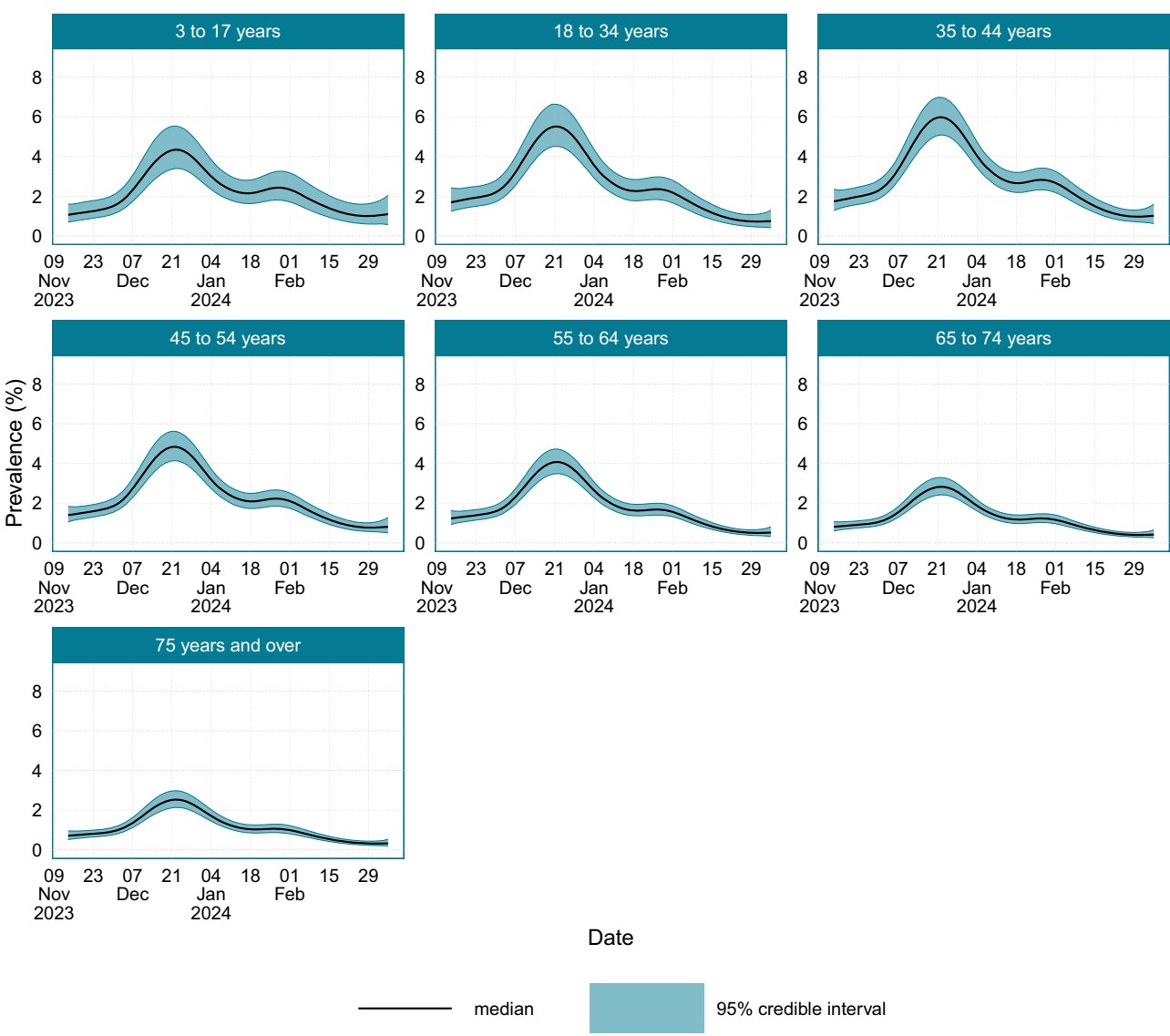

**Fig. 2 | The age-stratified prevalence of SARS-CoV-2 in England and Scotland between the 14th November 2023 and the 7th March 2024.** Source data are provided as a Source data file.

Scotland was approximately 0.06 day⁻¹, and the minimum test sensitivity occurred when the epidemic growth rate in England and Scotland was approximately −0.05 day⁻¹.

The age-stratified population average test sensitivity value is shown in Fig. 9, and a summary of the average, maximum and minimum population average test sensitivity is provided in Supplementary Table 6. The average test sensitivity for those aged 3–17 years was substantially lower than the other age groups at 62.0% (95% CI: 55.8–67.9). The average test sensitivity was broadly consistent across those aged 18–74 years, with those aged 75 years and over having the highest average test sensitivity at 79.2% (95% CI: 76.9–81.3). The location-stratified average test sensitivity over time is provided in Supplementary Fig. 11, with regions following the overall trend in England and Scotland closely. The average, minimum and maximum test sensitivity for each location is provided in Supplementary Table 7.

To understand the importance of accounting for time-varying LFD test sensitivity when modelling prevalence, the expected error incurred from not adjusting for time-varying test sensitivity is provided in Fig. 10. While the actual variations in the test sensitivity are relatively minor, as the reciprocal sensitivity is used to obtain prevalence from positivity, these small differences could result in potentially large errors. In those aged 3–17 years, where the average test sensitivity was lowest, the error incurred from using a constant estimate of test sensitivity exceeded 10% during the study. In those aged 75 years and over, where average test sensitivity is highest, the relative error in prevalence incurred from a constant estimate of test sensitivity occasionally exceeded 4%.

## Discussion

This study demonstrated that LFD tests, with the appropriate study design and statistical adjustment, can be used to develop robust estimates of incidence and prevalence. This results in a community prevalence study design with a substantially lower cost per test than earlier population-wide PCR-based studies, such as CIS and REACT, that were rapidly stood up in response to the public health emergency caused by SARS-CoV-2. As such, the design of this study is more suited to the current context where SARS-CoV-2 is no longer a public health emergency, though it continues to cause substantial morbidity and mortality.

Between the 14th November 2023 and the 7th March 2024, SARS-CoV-2 prevalence reached a peak value of 4.54% (95% CrI: 3.90–5.24%) in England and Scotland on 22nd December 2023. After the primary

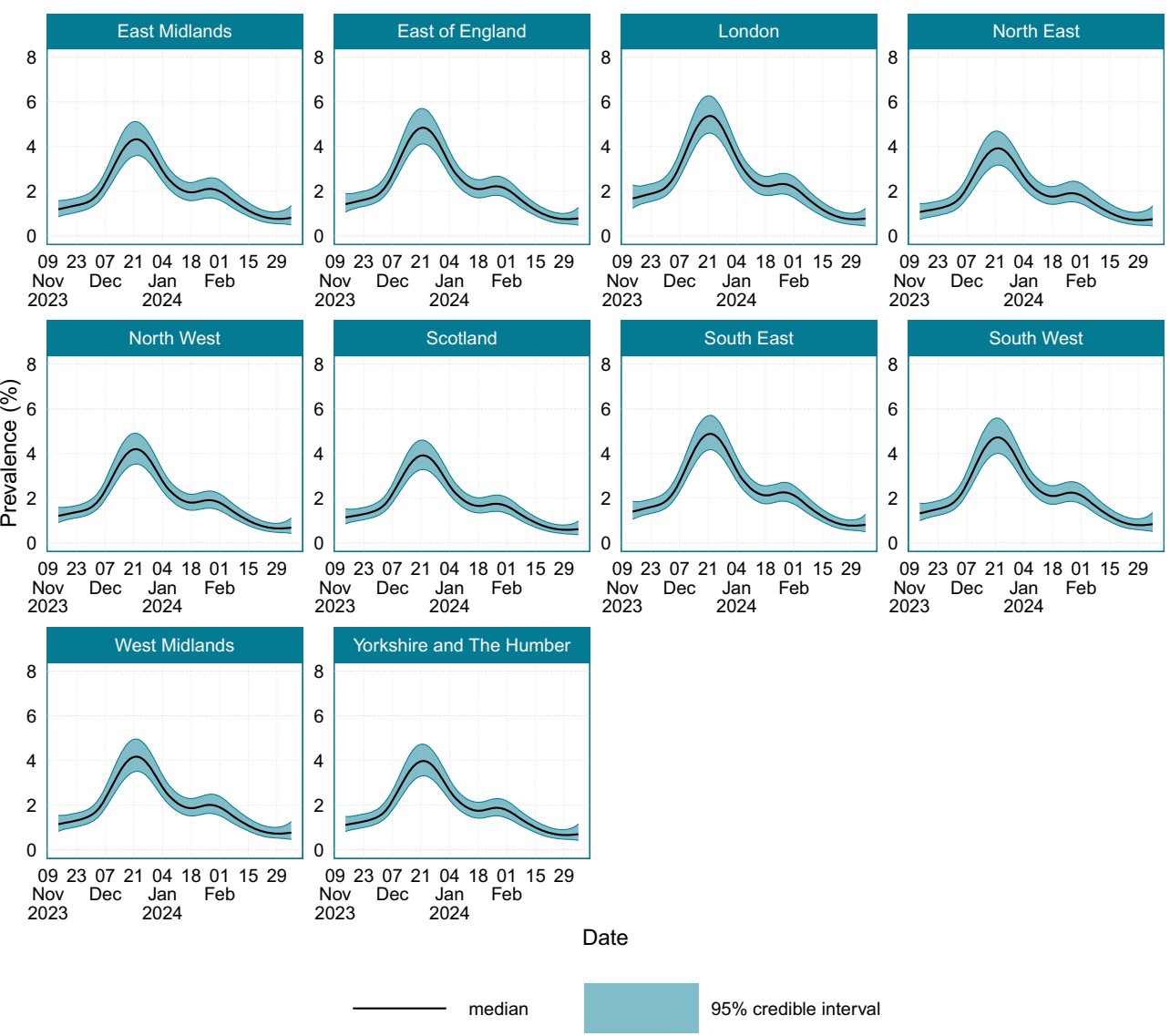

**Fig. 3 | The location-stratified prevalence of SARS-CoV-2 in England and Scotland between the 14th November 2023 and the 7th March 2024.** Source data are provided as a Source data file.

epidemic wave, there is some evidence of a short period of positive epidemic growth during mid-January 2024. This roughly coincides with the return to work and school after the winter holiday period in England and Scotland. The variant JN.1 grew rapidly in England and Scotland during the study period[13], which may have driven some of the epidemic wave. However, it is challenging to disaggregate the relative roles of variant pressure compared to expected increases in incidence due to seasonal mixing patterns.

There was limited spatial variation in the incidence, prevalence, and sensitivity. Though incidence/prevalence was slightly higher in London and slightly lower in the North East of England and Scotland. Previous research has demonstrated that there was substantial SARS-CoV-2 spatial heterogeneity early in the SARS-CoV-2 pandemic; however, this heterogeneity later disappeared, likely due to changes in mixing behaviours and shifting immunological dynamics[14]. Under the current levels of population mixing and given the highly infectious nature of SARS-CoV-2, limited spatial heterogeneity is expected. Other SARS-CoV-2 surveillance showed little spatial heterogeneity during the study period, though direct comparisons against other surveillance signals are difficult to make as they are often impacted by other sources of bias.

Differences in incidence/prevalence between the sexes were negligible. Prevalence and incidence varied across different age groups, with the highest incidence/prevalence occurring in young adults, and the lowest incidence/prevalence occurring in those aged 75 years or over. During the study period, an individual aged 35–44 years would have experienced, on average, 0.338 (95% CI: 0.288–0.395) infections, whereas an individual aged 75 years or over would have experienced, on average, 0.122 (95% CI: 0.103–0.143) infections. This likely results from the higher contact rates in younger age groups, leading to higher levels of transmission[15]. As part of a winter vaccination programme, individuals who either have a weakened immune system or were aged over 75 years would have been offered a vaccination during the Autumn, and this may have also contributed to the lower incidence and prevalence within those age groups.

The model structure used a convolution function approach to obtain the various quantities of interest as convolutions of the incidence. This enabled estimating a time-varying test sensitivity value that accounts for the recent epidemic dynamics and the age-dependent test sensitivity profiles that arise due to differing viral load trajectory characteristics across age groups. As such, the model

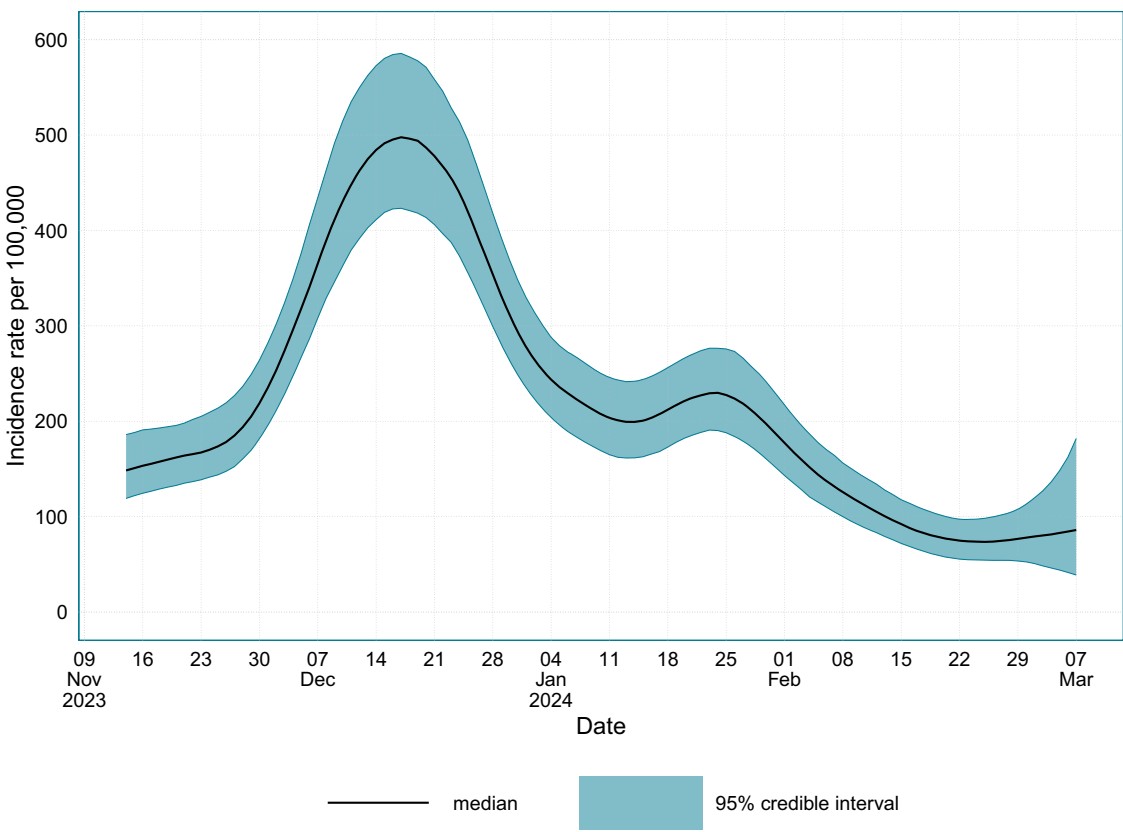

**Fig. 4 | The incidence of SARS-CoV-2 infections in England and Scotland from the 14th November 2023 to the 7th March 2024.** Source data are provided as a Source data file.

adjusts for the altered average LFD test sensitivity values during periods of rapid epidemic growth or decline and accounts for the different test sensitivity profiles across different age groups. During the study period, the average test sensitivity in England and Scotland was 72.1% (95% CI: 70.3–74.0), with variation between a maximum value of 77.2% (95% CI: 75.3–79.2) and a minimum value of 68.6% (95% CrI: 66.4–70.7). The average test sensitivity was lowest in the 3–17-year age group. Studies have previously demonstrated that younger individuals typically have lower viral load trajectory peaks and may also clear the virus faster[16]. As a result of the different viral load dynamics, the LFD test sensitivity profile estimated from the WCIS repeat-testing data[8] is lower for younger age groups, resulting in a lower average test sensitivity in younger age groups in this study. Overall, the average LFD test sensitivity values reported here are consistent with the results from Eyre et al.[11].

The results demonstrate that it is important to account for time-varying test sensitivity across age groups in LFD-based SARS-CoV-2 prevalence studies, as using a constant test sensitivity value would have incurred an absolute relative error in the estimated prevalence of up to 10% in the youngest age groups during this study. If time-varying test sensitivity is not accounted for, then prevalence would be over-estimated during periods of rapid epidemic growth, and under-estimated during periods of rapid epidemic decline. This is concerning, as periods of rapid epidemic growth/decline are when accurate surveillance is particularly important.

The exponential growth rate of the 23/24 winter SARS-CoV-2 wave peaked at around 0.06 day$^{-1}$; however, substantially higher growth rates were observed during the early pandemic[17,18]. Adjusting for the effect of epidemic phase bias on LFD test sensitivity is therefore an important consideration for any future prevalence studies that use LFD tests, particularly if the epidemic wave is growing rapidly. LFD-based prevalence studies for other pathogens may be more impacted by

epidemic phase bias in the test sensitivity due to the different viral load dynamics of those pathogens[19].

Multilevel regression and post-stratification was used to produce representative estimates. This was particularly important given that the study sample population was strongly biased towards older age groups, likely a consequence of study participants not being compensated, unlike the previous CIS study. Study designs where participants are volunteers allow for an increased sample size due to reduced costs per participant, when compared to compensated individuals, and therefore may be preferable if the non-representativeness of the sample can be fully accounted for. Despite the under-representation of some age groups, it was possible to produce estimates that parsimoniously explained the observed data for each stratum of the study population and adjusted for over-/under-representation in the sample.

## Limitations

As part of the study design, individuals were provided with a 10-day window to report their test result within; however, individuals who tested earlier in the window were observed to be more likely to return a positive result. Therefore, it was necessary to adjust for this to remove unintended daily effects induced by the overlapping test windows schedule. Likely, this was symptomatic individuals wanting to use their LFD test and waiting until the testing window was open. While this unintended behaviour was corrected for in the model, there remain open questions regarding the ideal length of the testing window. A shorter testing window may reduce study participation from individuals who do not currently have symptoms and therefore cause participation bias, whereas a longer testing window allows for modelling the testing window phenomenon.

Given the limited spatial variation, future SARS-CoV-2 community prevalence studies may consider reducing the amount of sampling required by focussing on obtaining a representative age-stratified

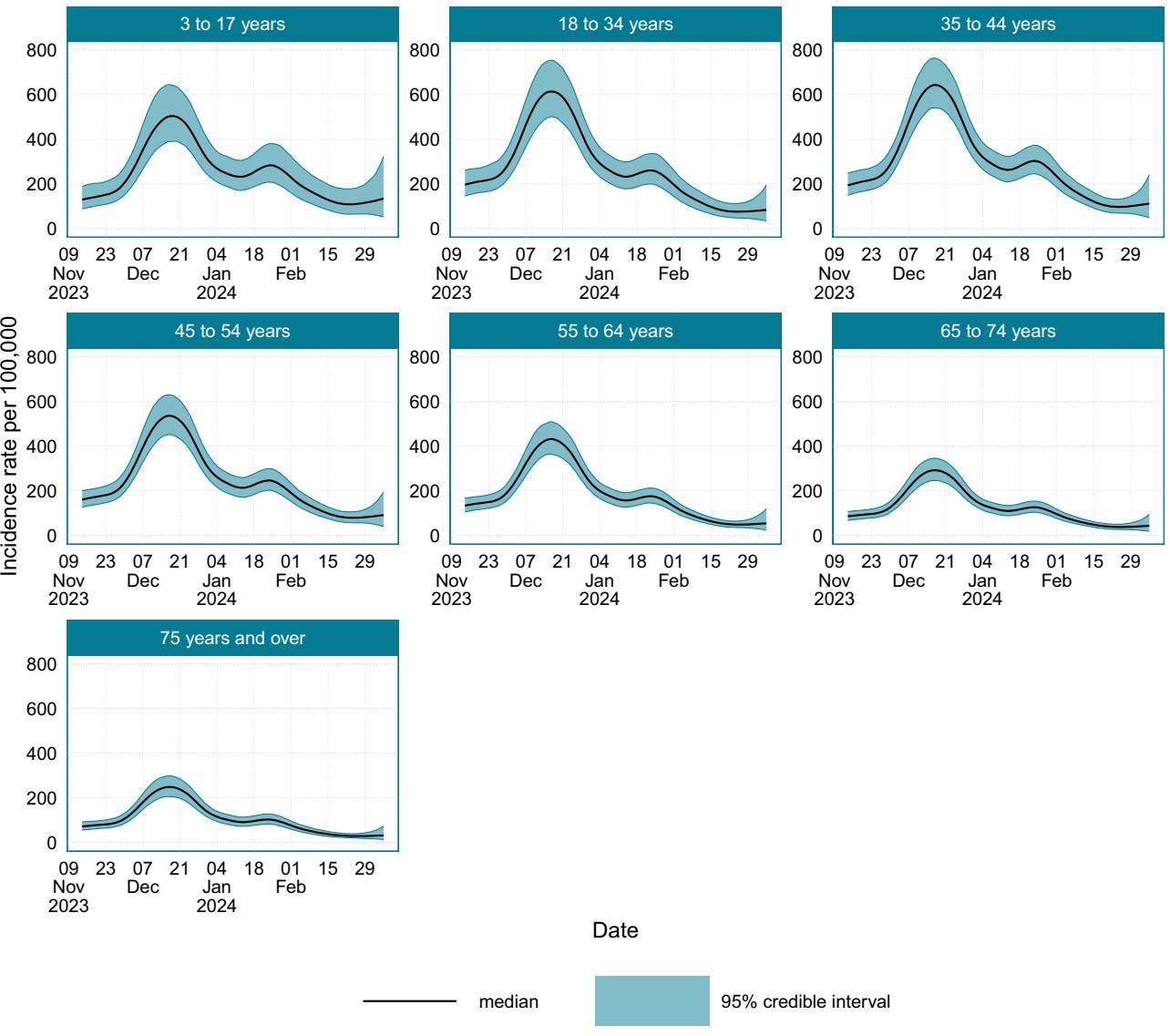

**Fig. 5 | The age-stratified incidence of SARS-CoV-2 infections in England and Scotland from the 14th November 2023 to the 7th March 2024.** Source data are provided as a Source data file.

sample, rather than a geographically stratified sample. Study participants were originally provided with 4 LFD tests as part of the main survey, and an additional 10 LFD tests for use in repeat testing. The repeat-testing regime, therefore, added substantial cost to the study; however, it was necessary to accurately estimate SARS-CoV-2 prevalence and incidence while adjusting for time-varying LFD test sensitivity. While participation in the main study was consistently high, uptake and completion of the repeat-testing regime were lower, and this area of the study design may benefit from further incentivisation, given that a large number of tests were provided for use in repeat testing.

A major limitation of the study is participation bias. Study participants generally had a high level of engagement with the testing; despite this, it is possible that participation bias arises due to the individuals who did not participate in the study. Of the 290,000 individuals invited, 139,453 individuals consented to participate in the study. Older age groups were over-represented, and females were slightly over-represented; however, the over-representation of both these groups was adjusted for in the post-stratification. Those reporting a white ethnicity were over-represented in the study sample, and it was not possible to adjust for this in the post-stratification. It is

possible that recent infection history, vaccination status, or other comorbidities or demographic variables may also affect participation in the study; however, it was not possible to assess this. Since the study was voluntary and participants did not receive remuneration, it is possible that the study will be biased towards more health-conscious individuals. This will increase the quality of the self-reported data, but the bias towards health-conscious individuals may not be representative of the whole population.

The study design of WCIS addressed several key challenges faced in previous SARS-CoV-2 prevalence studies and successfully demonstrated the use of LFD tests in respiratory pathogen community surveillance. LFD tests have a lower cost per unit when compared to PCR tests, which could make community prevalence studies more feasible in the current context, where the SARS-CoV-2 epidemic is no longer a public health emergency[20]. However, the use of LFD tests introduces several challenges to community prevalence studies, such as needing to accurately account for time-varying test sensitivity. To overcome these challenges, WCIS was designed to facilitate efficient parameter estimation and featured a modelling approach that accounts for the complex relationships between the different quantities of interest. Future community prevalence studies should explore the use of

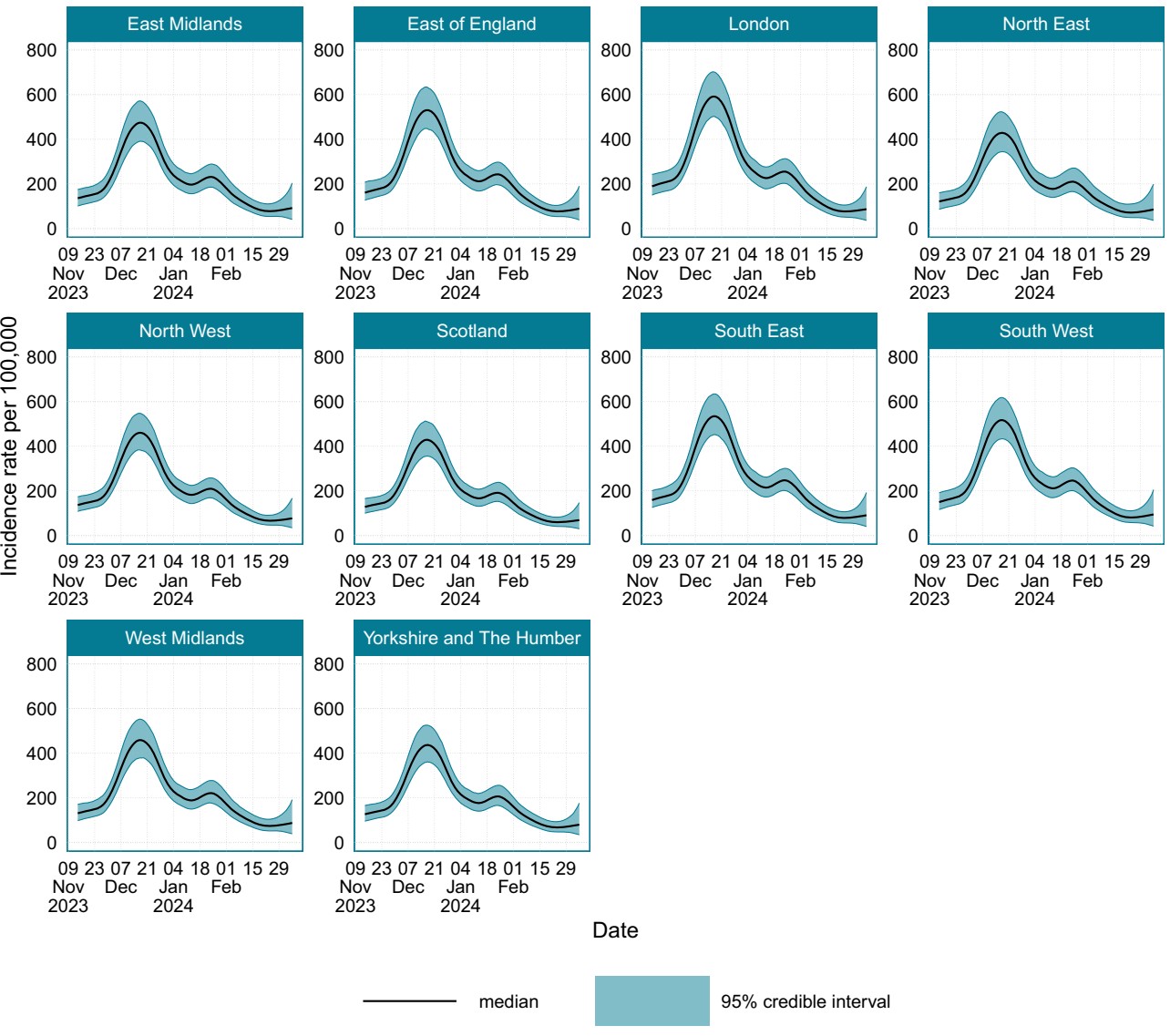

**Fig. 6 | The location-stratified incidence of SARS-CoV-2 infections in England and Scotland from the 14th November 2023 to the 7th March 2024.** Source data are provided as a Source data file.

multiplex LFD tests[19], which simultaneously test for a panel of different pathogens using a single swab. A community prevalence study design using the methods and design of WCIS with multiplex LFD tests could provide accurate surveillance of many different pathogens with a lower cost per test than the equivalent PCR test.

## Methods

### Epidemiological data

The cohort sampled for this study was selected from previous participants of the COVID-19 Infection Study (CIS)[21], a household study. Participants were asked if they would consent to being contacted by UKHSA for future studies—those who selected yes were asked to enrol.

If an individual agreed to participate, they were asked to read the participant information for this study and complete the study consent form. For participants aged under 16 years, a guardian was asked to read, share and discuss the participant information[22] for this study and complete the study consent form. Individuals eligible for the study lived in private households and were 3 years and older, living in England or Scotland. Participants were not offered financial incentives for the study, but were given the LFDs as part of the operations.

Testing data were collected as part of the WCIS. Every 4 weeks, study participants were given a 7-day window, within which they were asked to perform an LFD test and complete a short questionnaire. If the participants' LFD test was positive, they were asked to answer a follow-up questionnaire and participate in a repeat-testing regime, whereby they continued to take an LFD test every other day until testing negative for two consecutive tests. These data, including the initial test results, repeat tests and questionnaire results, were reported to ONS and then made available to UKHSA. Further details on the cohort, study design and data processing are provided in the supplementary materials. A breakdown of the sample size of each stratum is provided in Supplementary Table 1.

Demographic information about each individual surveyed is recorded alongside their testing results, including their age, sex, and location of residence in England and Scotland. The age groups considered for analysis are as follows: 3–17 years, 18–34 years, 35–44 years, 45–54 years, 55–64 years, 65–74 years, and 75 years and over. Locations are reported at a combination of Scotland and the English regions, giving ten total locations: East Midlands, East of England, London, North East, North West, Scotland, South East, South West, West Midlands, Yorkshire and the Humber.

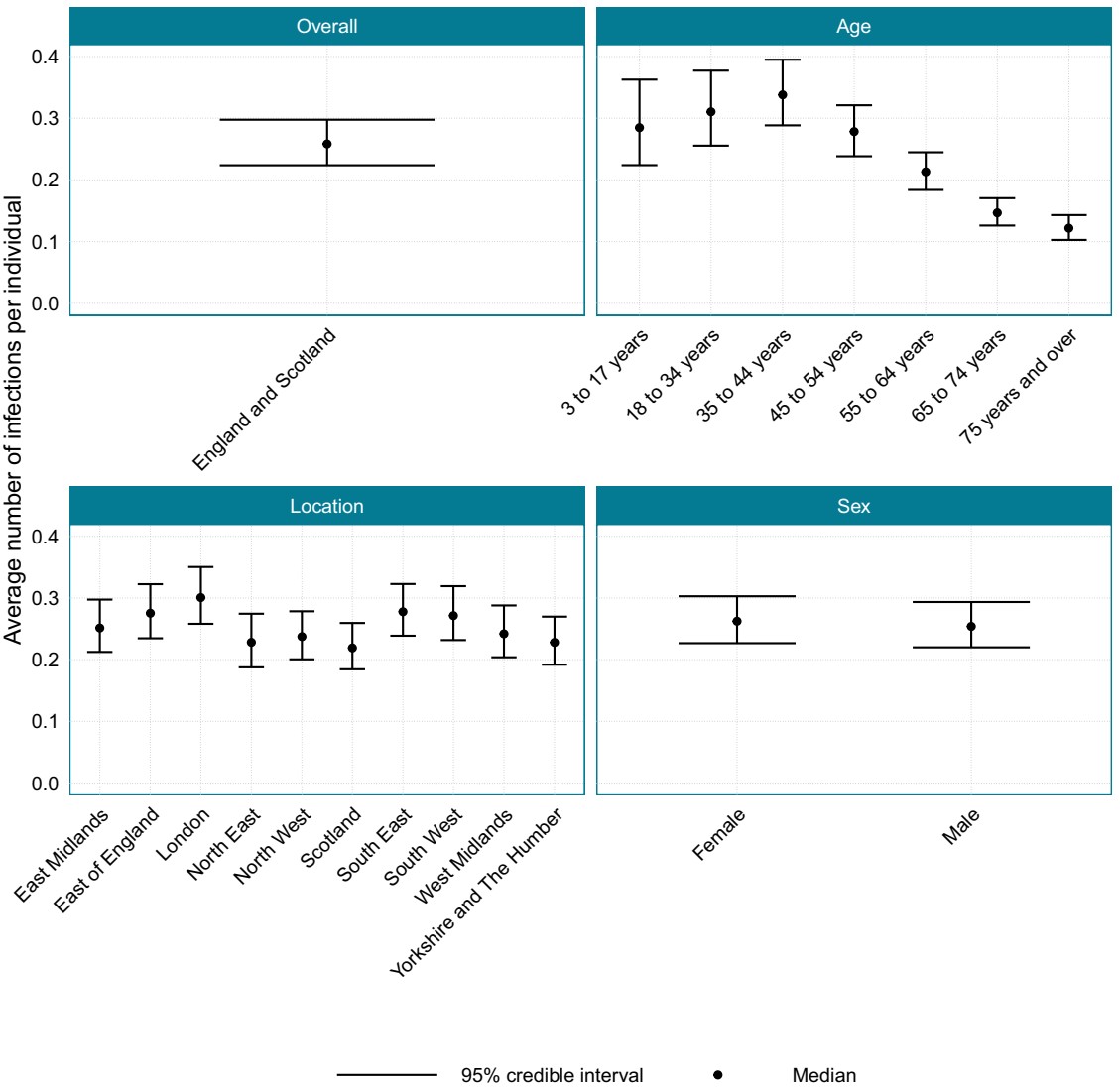

**Fig. 7 | The average number of infections that occurred per individual during the study period, 14th November 2023 to 7th March 2024, for England and Scotland, and when stratified by age, location and sex.** Source data are provided as a Source data file.

## Model structure

To begin, we briefly outline; the different datasets that are collected, the epidemiological parameters that need to be estimated, and the relationships between the key quantities of interest.

This study integrates epidemiological parameter models developed in our previous work[8], which used the repeat-testing data collected by WCIS. The definition of a currently infected individual is therefore inherited; an individual is defined as being currently infected if they have a non-zero probability of returning a positive LFD test result that is not a false positive. These individuals are referred to as being in the LFD-positive state. This definition is a natural consequence of the study being performed using only LFD tests, since there are no PCR tests to compare against.

During the study, approximately every 4 weeks, participants were asked to self-test using LFD tests, which produced a daily time series of LFD outcomes across different strata to inform sample positivity. Once a participant had tested positive, they were asked to re-test every other day until two consecutive negative test results were observed. In Overton et al.[8], the repeat-testing data were used to produce age-stratified estimates of the duration of LFD positivity in infected individuals and the LFD test sensitivity profile over the course of an

infection. Here, the estimated duration of LFD positivity and the LFD test sensitivity profile are then used to obtain both the positivity and prevalence time series as convolutions of the incidence time series, with the positivity then being fitted to the observed time series of LFD outcomes from the main survey. This approach ensures the complex epidemiological relationships between incidence, prevalence, and positivity are fully accounted for.

Study participants are provided with a 7-day window within which to perform their LFD test; however, in practice, participants were able to submit test results 2 days before the window opened, and 1 day after the window closed. Therefore, the effective window was 10 days in length, though submissions outside of the intended 7-day window were in the minority. Individuals who tested earlier in the window were observed to be more likely to test positive, and this window effect needed to be adjusted for when fitting to the observed positivity data.

The study population is stratified by 7 age groups, 10 locations, and sex, resulting in 140 different strata. Consequently, this can result in a small number of samples in each stratum each day. A multilevel regression approach is used to reduce the uncertainty in estimates by pooling information across strata. Post-stratification is then performed to obtain representative estimates across subgroups by

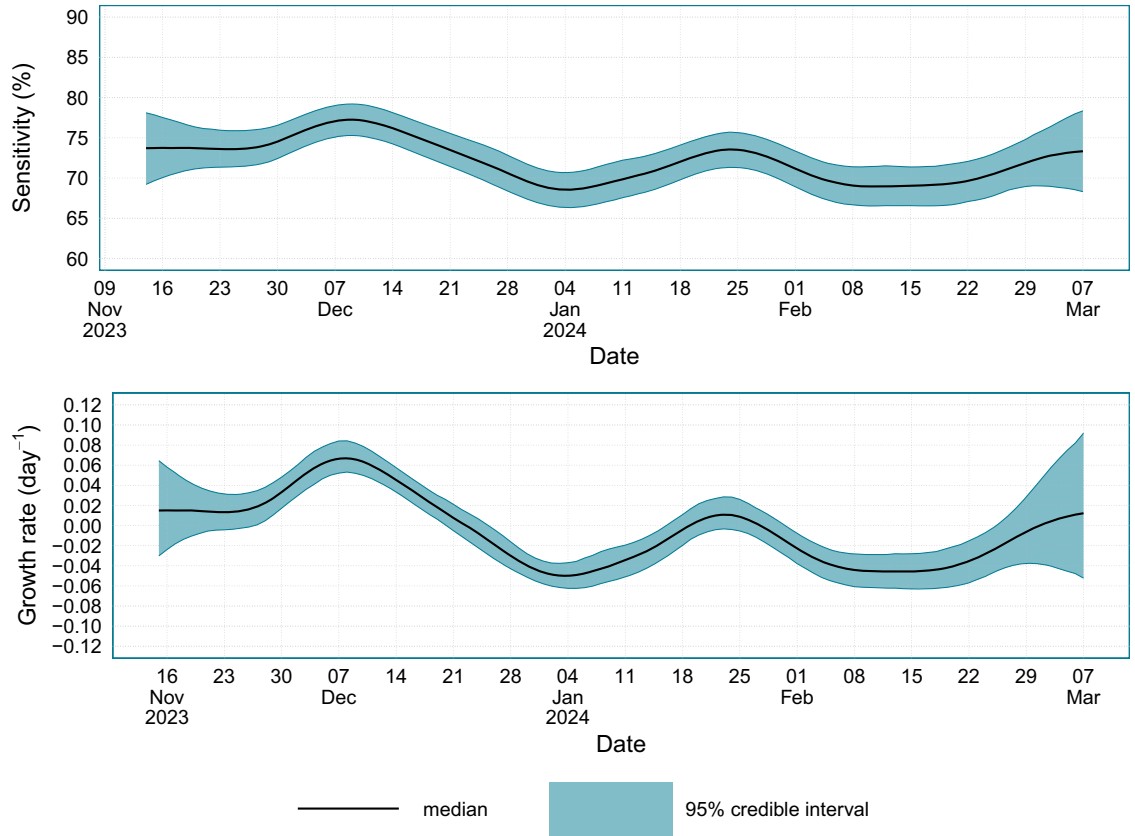

**Fig. 8 | Average LFD sensitivity alongside the epidemic growth rate.** Estimates of average LFD sensitivity and the incidence exponential growth rate for England and Scotland from the 14th November 2023 to the 7th March 2024. Source data are provided as a Source Data file.

adjusting for the over-/under-representation of different strata within the sample.

The time-varying average test sensitivity in a population is derived from the estimated positivity and prevalence time series, rather than being supplied as an input to the calculation of prevalence from positivity[10]. This provides an estimate of the test sensitivity that accounts for epidemic phase bias[23]. Epidemic phase bias is where the epidemic phase, e.g. growth or decline, biases the distribution of times since infection in infected individuals. When the epidemic is growing rapidly, the time since infection is biased towards smaller values, as there are expected to be more new infections than the previous day. This alters the expected test sensitivity value, as viral load typically peaks a few days after infection and therefore test sensitivity peaks a few days after infection. To assess the importance of estimating a time-varying test sensitivity model, a counterfactual analysis is performed to estimate the error had a constant estimate of test sensitivity been used to estimate prevalence from positivity.

**Convolution function approach.** Any time series of an epidemic quantity, $Q(t)$, can be obtained as the convolution of the incidence time series $I(t')$ up until time $t$, and a function $f_Q(\tau)$ that describes the relationship between the incidence time series and the quantity of interest[24];

$$Q(t) = \int_0^\infty I(t - \tau) f_Q(\tau) d\tau \qquad (1)$$

The model, therefore, begins with the incidence time series and convolves the incidence with different functions that describe SARS-CoV-2 epidemiology to obtain a time series of expected positivity and prevalence.

Since SARS-CoV-2 infections cannot remain positive indefinitely, for both the prevalence and positivity time series, it is assumed there is a maximum length of time that individuals can remain infected for, $W \in \mathbb{R}_+$. This implies that the current prevalence and positivity are effectively only conditional upon the previous $W$ days of incidence, i.e. for $\tau > W$ we have that $f_Q(\tau) = 0$. Therefore, the indefinite integral can be approximated as

$$Q(t) = \int_0^\infty I(t - \tau) f_Q(\tau) d\tau \approx \int_0^W I(t - \tau) f_Q(\tau) d\tau \qquad (2)$$

Note that $Q(0)$ depends on $t - \tau < 0$, which implies that all epidemic quantity time series depend on values of the incidence time series that occurred prior to the start of the study. To compute $Q(t)$, the incidence time series is modelled on the interval $[-W, T]$, where $T$ is the last time point in the study, and the positivity and prevalence time series are modelled on the interval $[0, T]$. The interval $[-W, 0)$ is referred to as the warmup interval and is weakly identified. Consequently, while the warmup interval is necessary for model fitting, it does not have a clear interpretation and is not presented in the results section of the paper. For this paper, a value of $W = 20$ was used, as the probability of an individual remaining LFD positive for more than 20 days is small[8].

Let $I_{a,l,s}(t) \in [0,1]$ denote the incidence time series for a given age, location, and sex combination, where incidence is defined as the proportion of that subgroup newly infected on day $t$. The prevalence time series, $\text{Prev}_{a,l,s}(t) \in [0,1]$, is defined as the proportion of that subpopulation that is in the LFD-positive state on day $t$. Let $f_{\text{prev}}^a(\tau)$ be the probability that an individual in age group $a$ is in the LFD-positive state $\tau \in \mathbb{R}_+$ days after they were infected. The prevalence time series

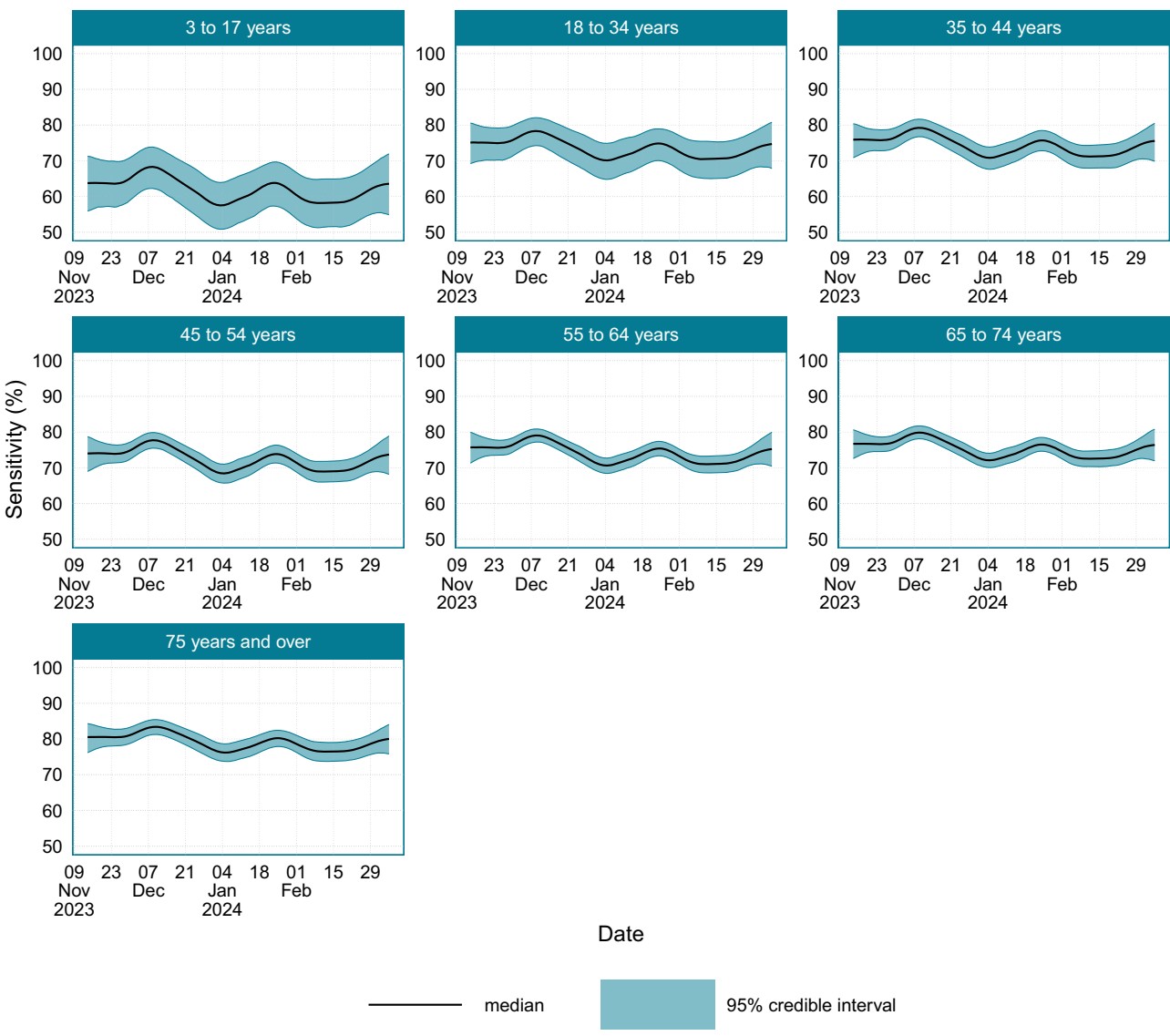

**Fig. 9 | The age-stratified LFD test sensitivity in England and Scotland from the 14th November 2023 to the 7th March 2024.** Source data are provided as a Source data file.

is then obtained by convolving $f_{\text{prev}}^a(\tau)$ with the incidence time series;

$$\text{Prev}_{a,l,s}(t) = \int_0^W I_{a,l,s}(t-\tau) f_{\text{prev}}^a(\tau)\, d\tau \qquad (3)$$

To calculate the expected sample positivity, it is necessary to account for prevalence, sensitivity, and specificity. The false positive probability and the true positive probability are analysed separately via

$$\text{Pos}_{a,l,s}(t) = \text{Pos}_{a,l,s}^{\text{TP}}(t) + \text{Pos}_{a,l,s}^{\text{FP}}(t) \qquad (4)$$

where the superscript TP and FP represent true and false positives, respectively. The true positive probability is given by the probability that a randomly sampled member of the stratum is in the LFD-positive state and, when tested, returns a true positive result. The false positive probability is the probability that an uninfected member of the population is tested and returns a false positive test result.

In Overton et al.[8], a model is developed to describe the probability of observing a true positive LFD test given that an individual in age group $a$ is still LFD-positive $\tau$ days after they were infected, denoted by $f_{+|+}^a(\tau) \in [0,1]$. This is referred to as the test sensitivity profile of LFD-positive individuals. Individuals who were recently infected are likely

to have high viral loads, and therefore, the LFD test sensitivity is high for small values of $\tau$. As $\tau$ increases, the test sensitivity decreases as individuals who remain LFD-positive approach the end of their infection, which implies a decreasing viral load. Test sensitivity is highly dependent upon the viral load of an individual, and it has been shown that there are differences in the viral load trajectory across age groups, with younger individuals having lower viral load peaks. The effects of these differing viral load dynamics across age groups are observed in the test sensitivity profiles estimated by Overton et al., with individuals aged 3–17 years having a lower LFD test sensitivity than other age groups.

The integrand for the prevalence convolution, $I_{a,l,s}(t-\tau) f_{\text{prev}}^a(\tau)$, provides the proportion of the population that were infected at time $t-\tau$ and were still LFD-positive $\tau$ days later at time $t$. For the positivity convolution, $I_{a,l,s}(t-\tau) f_{\text{prev}}^a(\tau) f_{+|+}^a(\tau)$ provides the probability that a randomly tested member of the subpopulation at time $t$ was infected at time $t-\tau$, is still LFD-positive $\tau$ days later and returns a true positive test result. Therefore, the true positive probability at time $t$ is given by

$$\text{Pos}_{a,l,s}^{\text{TP}}(t) = \int_0^W I_{a,l,s}(t-\tau) f_{\text{prev}}^a(\tau) f_{+|+}^a(\tau)\, d\tau \qquad (5)$$

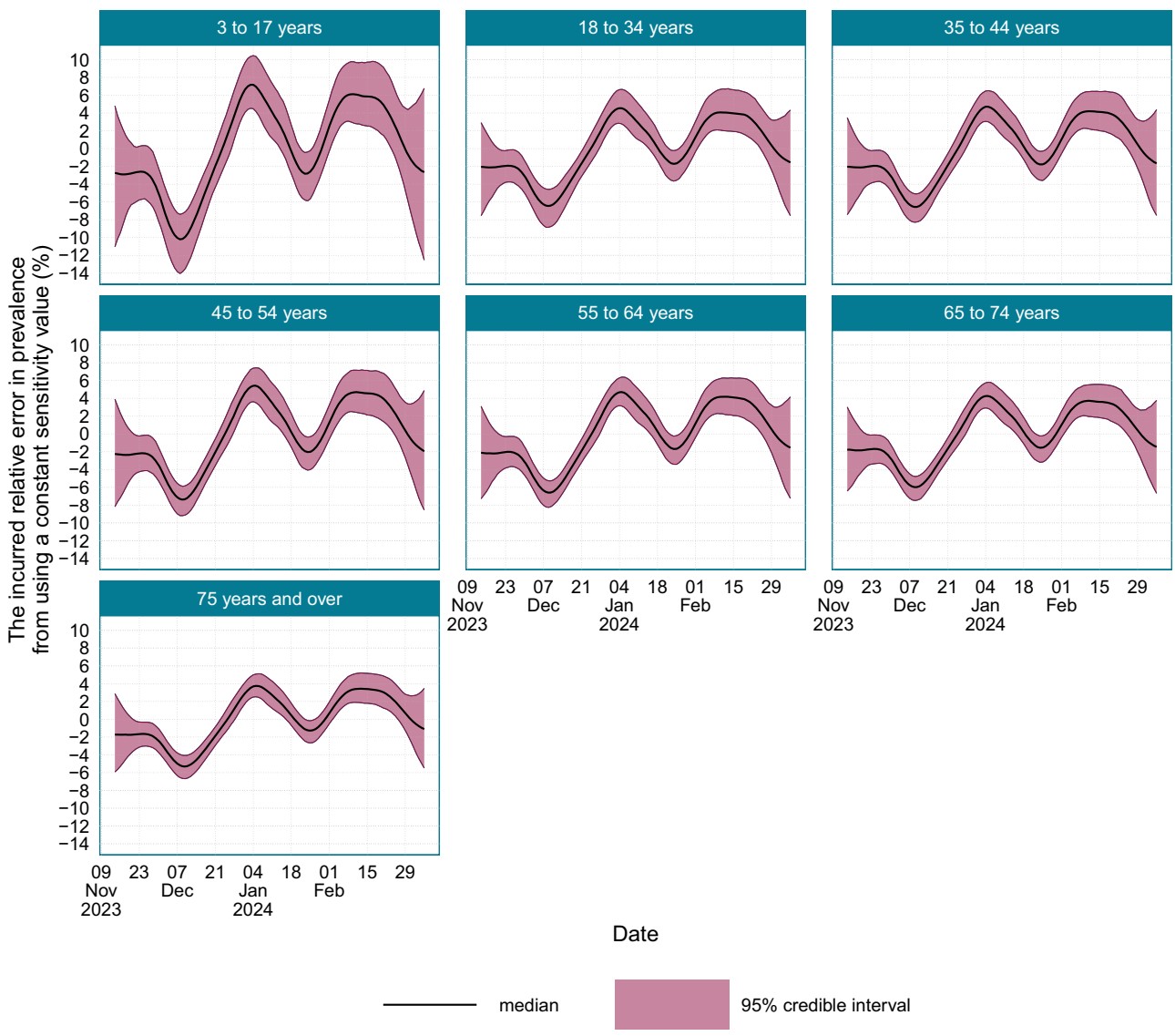

**Fig. 10 | The age-stratified relative error in the prevalence estimates if time-varying sensitivity had not been accounted for.** Source data are provided as a Source data file.

Under the assumption that an uninfected individual returns a false positive test result independently at random with probability $1 - \gamma \in [0, 1]$, where $\gamma$ is the test specificity, the time-varying false positive probability is given by

$$\text{Pos}^{\text{FP}}_{a,l,s}(t) = (1 - \text{Prev}_{a,l,s}(t))(1 - \gamma) \tag{6}$$

Combining this together, the positivity time series is given by

$$\text{Pos}_{a,l,s}(t) = \int_0^W I_{a,l,s}(t - \tau) f^a_{\text{prev}}(\tau) f^a_{+|+}(\tau) \, d\tau + (1 - \text{Prev}_{a,l,s}(t))(1 - \gamma) \tag{7}$$

Given that $1 \approx (1 - \text{Prev}_{a,l,s}(t))$ and $(1 - \gamma) \approx 10^{-4}$, for computational efficiency, the false positive rate is approximated as

$$\text{Pos}^{\text{FP}}_{a,l,s}(t) \approx (1 - \gamma) \tag{8}$$

**Likelihood and model fitting.** Additional effects are modelled to account for individuals' propensity to test at different points in their assigned testing window. Participants were observed to be more likely to test positive if they tested at the very start or prior to their window

opening. The window effects are assigned a weakly-informative prior of

$$\beta^{\text{window}}_w \sim \mathcal{N}\left(0, \sqrt{15}\right)$$

where $w \in \{-2, -1, \ldots, 7\}$ is the number of days since the testing window opened. These testing effects are constrained so that the average effect on the prevalence time series is zero over the course of the testing window when weighted by the number of tests, $n_i$, taken on each window index, $i$:

$$\sum_{i=-2}^7 n_i \beta^{\text{window}}_i = 0 \tag{9}$$

Typically, this constraint would be implemented using a simplex; however, for computational efficiency, we implement this as a soft constraint using

$$\sum_{i=-2}^7 n_i \beta^{\text{window}}_i \sim \mathcal{N}\left(0, 10^{-3}\right) \tag{10}$$

The expected rate of positivity after accounting for window effects is then calculated as:

$$p_{a,l,s,w}(t) = \text{logit}^{-1}\left(\text{logit}\left(\text{Pos}_{a,l,s}(t)\right) + \beta_w\right) \tag{11}$$

The observed series of positive tests each day and in each of the subgroups, $y_{a,l,s,w}(t)$, is modelled using a binomial distribution with a probability of the predicted rate of positivity for each subgroup at each time point

$$y_{a,l,s,w}(t) \sim \text{Binomial}\left(N_{a,l,s,w}(t), p_{a,l,s,w}(t)\right) \tag{12}$$

The model generates posterior samples of the most credible incidence time series, prevalence time series, and epidemiological parameters that explain the observed data for each combination of age, location and sex.

**Multilevel regression.** A Bayesian multilevel regression model is used to estimate incidence for each demographic stratum. A national-level trend of log-odds incidence,

$$f^{\text{nat}}(t) \in \mathbb{R}^{T+W} \tag{13}$$

is combined with per-age, per-location, and per-sex deviations from the national-level trend, each of which are also allowed to vary over time;

$$f_l^{\text{loc}}(t) \in \mathbb{R}^{T+W} \forall l \in L \tag{14}$$

$$f_a^{\text{age}}(t) \in \mathbb{R}^{T+W} \forall a \in A \tag{15}$$

$$f^{\text{sex}}(t) \in \mathbb{R}^{T+W} \tag{16}$$

Some values of the incidence time series and duration of positivity distributions will result in prevalence values that are greater than 1. These incidence time series would be impossible to observe, e.g. 75% of a population being infected 2 days in a row, leading to an impossible 150% prevalence. However, during the MCMC initialisation and adaptation steps, these invalid incidence time series values can result in many draws being rejected, which causes poor adaptation of the MCMC sampler and results in inefficient sampling. To prevent this, the incidence is restricted such that $I_{a,l,s}(t) \in [0, 0.05]$, implying that no more than 5% of any subgroup population is infected per day. SARS-CoV-2 has an average duration of LFD positivity of approximately 10 days[8], and therefore the upper limit of 5% incidence per day places an upper limit of approximately 50% prevalence—significantly higher than any reliable estimate of SARS-CoV-2 prevalence in a large population. Further, current epidemiological dynamics, such as susceptible depletion and acquired immunity from infection or vaccination, likely make it effectively impossible for such a high incidence/prevalence to be achieved. Therefore, setting the incidence upper limit at 5% is expected to only remove biologically implausible incidence time series values, while allowing the model to initialise efficiently. When the model was fitted, the estimated incidence values did not approach the 5% upper limit, implying the model was not constrained by this upper limit.

Letting $s$ take values of either $-1$ or $1$ we have

$$I_{a,l,s}(t) = 0.05 \cdot \text{logit}^{-1}\left(f^{\text{nat}}(t) + f_l^{\text{loc}}(t) + f_a^{\text{age}}(t) + sf^{\text{sex}}(t)\right) \tag{17}$$

Each of the national, location, age and sex-level trends is assigned second-order random-walk (RW2) smoothing priors[25,26] as follows:

$$f^{\text{nat}}(t) \sim \text{RW2}\left(\mu^{\text{nat}}, \tau^{\text{nat}}\right) \tag{18}$$

$$f_l^{\text{loc}}(t) \sim \text{RW2}\left(\mu_l^{\text{loc}}, \tau^{\text{loc}}\right) \forall l \in L \tag{19}$$

$$f_a^{\text{age}}(t) \sim \text{RW2}\left(\mu_a^{\text{age}}, \tau^{\text{age}}\right) \forall a \in A \tag{20}$$

$$f^{\text{sex}}(t) \sim \text{RW2}\left(\mu^{\text{sex}}, \tau^{\text{sex}}\right) \tag{21}$$

The $\tau \in \mathbb{R}_+$ parameters control the smoothness of the time series, such that a smaller value for $\tau$ indicates more smoothing, and each $\mu \in \mathbb{R}$ parameter indicates the starting value of the random walk. The following prior for smoothness was used,

$$\tau^{\text{nat}}, \tau^{\text{loc}}, \tau^{\text{age}}, \tau^{\text{sex}} \sim \text{Exponential}(100) \tag{22}$$

The intercepts $\mu$ for each location, age group and sex are given zero-mean priors with pooled variances across all locations, age groups, and sexes:

$$\mu_l^{\text{loc}} \sim \mathcal{N}\left(0, \sigma^{\text{loc}}\right) \tag{23}$$

$$\mu_a^{\text{age}} \sim \mathcal{N}\left(0, \sigma^{\text{age}}\right) \tag{24}$$

$$\mu^{\text{sex}} \sim \mathcal{N}\left(0, \sigma^{\text{sex}}\right) \tag{25}$$

$$\sigma^{\text{loc}}, \sigma^{\text{age}}, \sigma^{\text{sex}} \sim \text{Exponential}(100) \tag{26}$$

For the national-level trend intercept, we assign the prior of

$$\mu^{\text{nat}} \sim \mathcal{N}(-4, 3) \tag{27}$$

The median of this prior gives an incidence rate of 90 new infections per day per 100,000 individuals, and the interval of (0.26, 4300) new infections per day per 100,000 individuals contains 95% of the prior probability mass. As such, the median value of this prior is on the right scale for the incidence rate; however, there is substantial flexibility for the model to explore if necessary.

After model fitting, post-stratification[27,28] is performed to produce estimates of the incidence and prevalence that are representative of a given demographic strata, e.g. those aged between 3 and 17 years, by adjusting for the over/under-representation of different strata within a given stratum. The size of each stratum is obtained using ONS household population estimate projections[29].

**Infections per individual.** The size of an epidemic wave can be contextualised by reporting the expected number of SARS-CoV-2 infections an individual will experience over the course of the study. Let $\lambda_{a,l,s}(t) \in \mathbb{R}_+$ be the total number of infections that occurred between day 0 and day $t$ of the study for a given stratum, given by

$$\lambda_{a,l,s}(t) = P_{a,l,s} \cdot \int_0^t I_{a,l,s}(\tau) \, d\tau \tag{28}$$

where $P_{a,l,s} \in \mathbb{R}_+$ is the stratum's population size. The average number of infections that occurred per individual between day 0 and $t$ of the study, $\alpha_{a,l,s}(t) \in \mathbb{R}_+$, is therefore given by

$$\alpha_{a,l,s}(t) = \frac{\lambda_{a,l,s}(t)}{P_{a,l,s}} = \int_0^t I_{a,l,s}(\tau) \, d\tau \tag{29}$$

It is possible for individuals to be infected more than once, and consequently, if the study had run for long enough, the average number of infections per individual would eventually exceed 1. If the

average number of infections per individual was equal to 1, this would not imply that each member of the population had been infected once, as some individuals would have been infected more than once, while others may not have been infected at all. Due to the length of the study, it is unlikely that a substantial number of individuals were infected more than once.

**Analysis of time-varying test sensitivity.** The population average test sensitivity, $\text{Sens}_{a,l,s}(t) \in [0,1]$, is defined as the probability of returning a positive test from a randomly selected LFD-positive member of a stratum at a given point in time. This differs from the test sensitivity profile of LFD-positive cases, which refers to the probability of returning a true positive test result given the time since infection of the LFD-positive individual. The population average test sensitivity is determined by the test sensitivity profile and the infectious age distribution, i.e. the distribution of times since infection in LFD positive individuals, of that population at that point in time.

For a given stratum, the population average test sensitivity can be expressed in terms of the prevalence and the expected true positive rate,

$$\text{Sens}_{a,l,s}(t) = \frac{\text{Pos}^{\text{TP}}_{a,l,s}(t)}{\text{Prev}_{a,l,s}(t)} \qquad (30)$$

Consequently, $\text{Sens}_{a,l,s}(t)$ is conditional upon the recent values of the incidence time series, weighted by the probability that infections several days ago remain positive and would return a false negative result. This results in a model of test sensitivity that is affected by epidemic phase bias, as it accounts for how the recent epidemic dynamics have impacted the infectious age distribution and the epidemiological characteristics of SARS-CoV-2 infections within that population.

For LFD tests, the magnitude of the effect that epidemic phase bias has on the test sensitivity is unknown. Here, we develop an expression for the error incurred had a constant value of test sensitivity been used, as this would not account for epidemic phase bias. Let $\overline{\text{Sens}}_{a,s,l}$ be the average of value of $\text{Sens}_{a,l,s}(t)$. The estimate of prevalence obtained using the average value of sensitivity is given by

$$\widehat{\text{Prev}}_{a,l,s}(t) = \frac{\text{Pos}_{a,l,s}(t)}{\overline{\text{Sens}}_{a,l,s}} \qquad (31)$$

The relative error in the prevalence at a given point in time from using a constant test sensitivity estimate is then calculated as

$$\text{Error}_{a,l,s}(t) = \frac{\text{Prev}_{a,l,s}(t) - \widehat{\text{Prev}}_{a,l,s}(t)}{\text{Prev}_{a,l,s}(t)} = 1 - \frac{\text{Sens}_{a,l,s}(t)}{\overline{\text{Sens}}_{a,l,s}} \qquad (32)$$

As the time-varying sensitivity is conditional on the recent epidemic dynamics, the error in the prevalence estimate is highly related to the exponential growth rate of the epidemic for each stratum, $r_{a,l,s}(t) \in \mathbb{R}^T$. Under the assumptions of an exponential growth model, we have that

$$I_{a,s,l}(t) = I_{a,s,l}(t-1) \cdot e^{r_{a,l,s}(t)} \qquad (33)$$

which allows the daily exponential growth rate to be estimated as

$$r_{a,l,s}(t) = \log(I_{a,l,s}(t)) - \log(I_{a,l,s}(t-1)) \qquad (34)$$

**Implementation.** The Bayesian model is implemented using Stan[30], a probabilistic programming language for Bayesian inference. The parameter estimation sub-models from Overton et al.[8], and the methodology described here are implemented in a single Stan programme. This approach ensures that uncertainty is properly propagated throughout the model. Model fitting was performed using Hamiltonian MCMC, with 8 chains using 500 warmup iterations and

500 sampling draws each. Model fitting took approximately 12 h to complete using 8 CPU cores.

Data wrangling and results analysis/plotting were performed in R version 4.3.2. The Stan model was fitted using Stan version 2.32.2 and cmdstanr version 0.8.1. Data wrangling and visualisation were performed using tidyverse 2.0.0.

## Ethics approval
The study received ethical approval from the National Statistician's Data Ethics Advisory Committee.

## Reporting summary
Further information on research design is available in the Nature Portfolio Reporting Summary linked to this article.

## Data availability
UKHSA operates a robust governance process for applying to access protected data that considers: the benefits and risks of how the data will be used compliance with policy, regulatory and ethical obligations data minimisation how the confidentiality, integrity, and availability will be maintained retention, archival, and disposal requirements best practice for protecting data, including the application of 'privacy by design and by default', emerging privacy conserving technologies and contractual controls. Access to protected data is always strictly controlled using legally binding data sharing contracts. UKHSA welcomes data applications from organisations looking to use protected data for public health purposes. To request an application pack or discuss a request for UKHSA data you would like to submit, contact DataAccess@ukhsa.gov.uk. Source data for figures are provided with this paper. Source data are provided with this paper.

## Code availability
Stan and R codes to reproduce the analysis on a synthetic dataset are under MIT license on GitHub at https://github.com/MFyles/WCIS-incidence-prevalence (https://zenodo.org/records/17209309[31]).

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

## Acknowledgements

The authors would like to thank the UKHSA Surveillance and Immunity team and the ONS WCIS analysis team for their contributions.

## Author contributions

All authors contributed extensively to the work presented in this paper. T.W. conceptualised the study. T.W. and J.M. supervised the study. M.F., T.W., R.P., C.O. and A.P. designed the methodology and performed the analysis. M.F., R.P., C.O., A.P., A.G., T.W. and J.M. developed the code. J.M. coordinated and managed the project. MF drafted the manuscript, and M.F., J.M., T.W., R.P., A.P. and C.O. reviewed and edited the manuscript.

## Competing interests

The authors declare no competing interests.
