## [Transparent Peer Review file · Nature Communications]

Estimating COVID-19 incidence and prevalence using lateral flow tests in England and Scotland, 2023-2024

Corresponding Author: Dr Martyn Fyles

Version 0:

Reviewer comments:

Reviewer #1

(Remarks to the Author)

Thank you for the opportunity to review this manuscript, which reads really well.

The authors have well contextualized the rationale for the proposed methodology. I have particularly enjoyed the clarity of the approach including all model assumptions and parameters.

While this work presents interesting yet not novel results, the framework and methods are noteworthy due to their contribution towards robust estimation of key indicators of current circulation of SARS-CoV-2 at the national level even in presence of weak laboratory testing.

Point 1:

I would encourage for example that the authors recognize more clearly that LFD tests can miss positive tests (unacceptable false negative rates) and refer to this work <https://pmc.ncbi.nlm.nih.gov/articles/PMC8758200/>

Point 2:

Ct values, a proxy for viral load, have been shown to have different characteristics by age. For example, evidence points largely to lower viral load in young children (i.e. higher Ct values). This raises the question on how to account for viral load dynamics in the age-stratified results. The authors have briefly mentioned this on Page 21 in the discussion but I would want to see some sensitivity analysis to lend credibility.

Point 3:

The limitations associated with participation bias is problematic. This deserves more discussion on whether the incidence, prevalence, and positivity in the older age group is truly representative amongst this group.

Point 4:

Limited spatial prevalence is a concern. The authors do take note that the literature shows significant heterogeneity in COVID-19 with respect to spatial factors. I would like see the incorporation of more spatial elements in the data (e.g. mobility) to account for the shift in dynamics

Point 5:

With respect to the multilevel regression, the authors pool age, sex, location using an additive logit function and the incidence is restricted to help the chain converge. How was the 5% threshold decided? Conversely, would MCMC model hold when incidence is very low given additive and independent nature of the inverse logit?

Point 6:

Compared to developing a Bayesian multilevel regression with time series convolution to capture repeated LFD testing with all the issues related to selection bias, age-dependent results, and absence of spatial variations, would this effort not be better suited to utilize wastewater-based surveillance data?

(Remarks on code availability)

Reviewer #2

(Remarks to the Author)

This study presents a valuable framework for assessing the incidence and prevalence of SARS-CoV-2 over a four-month period, coinciding with the circulation and dominance of the JN.1 variant in the UK. It uses a rigorous and well-structured methodology that accounts for potential biases associated with lateral flow device testing and sample collection timing. My evaluation focuses on the epidemiological aspects of the study, as my expertise is limited in the mathematical modeling domain. I have provided several suggestions that, in my view, could further strengthen the study's contribution to the existing literature.

- 1) The manuscript includes considerable technical statistical descriptions that may limit readability and accessibility for a broader audience, including policymakers and public health officials. I recommend summarizing key statistical methods and concepts using simpler terms within the methods section, while providing the detailed descriptions in an appendix.
- 2) It is unclear whether prior infection, vaccination history, and recent vaccination were factored into the analysis and to what extent these factors contributed to the observed decline in incidence and prevalence among older individuals, beyond differences in contact rates. Given that older adults may have had higher comorbidity rates and were likely prioritized for vaccination, these factors could have influenced their lower incidence and prevalence. Clarifying how these variables were accounted for would strengthen the epidemiological interpretation of the findings.
- 3) Please specify the extent of non-compliance with repeat testing and whether this varied across age groups, as well as the implications on the study findings.
- 4) It is unclear whether repeat testing participation differed between individuals who tested positive in the first round versus those who tested negative, and how any potential bias arising from this was addressed. If possible, providing a comparison of compliant versus non-compliant individuals in terms of repeat testing, stratified by demographic characteristics, would be valuable. More importantly, assessing differences in vaccination status, comorbidity profile, and prior infection history could help determine whether these factors influenced follow-up testing participation and the overall representativeness of the findings.
- 5) It would be helpful to clarify whether there were specific criteria for classifying a new positive test in a subsequent wave as a new incident infection. This is particularly relevant in differentiating between reinfections and prolonged viral shedding.

(Remarks on code availability)

The code is detailed and robust

Version 1:

Reviewer comments:

Reviewer #1

(Remarks to the Author)

Thank you to the authors for their careful attention to the points raised in the previous assessment of their submitted manuscript. This work provides an important methodological framework that addresses current limitations associated with LFD-based prevalence studies and provides a reproducible approach that could be replicated in the future across various geographies. I would like to note that I am unable to find the code nor README file.

1- With respect to the first point about the sensitivity of LFD tests, I agree that the reference provided is for whole population screening. Yet, the premise of the current work is at the population level rather than a specific subgroup (e.g. occupational, remote/rural areas). Therefore, I think that positioning the paper as a contribution to the field of population monitoring, screening, surveillance should metrics be LFD-based is important and adding a reference that speaks to the limitation of LFD tests is critical.

2- The discussions on pages 21 and 22 in relation to age-dependent sensitivity of the testing as well as spatial heterogeneity are very well articulated, thank you.

3- The impact of this paper would be much greater if the code was also available in a GitHub repository

(Remarks on code availability)

Reviewer #2

(Remarks to the Author)

The authors have addressed all of my comments. I have no further comments on this manuscript.

(Remarks on code availability)

|No comments on the code

Response to reviewers:

We sincerely appreciate the time and effort that the reviewers dedicated to reviewing our manuscript. Their comments have been invaluable in helping us strengthen our work. We have addressed their comments below.

Reviewer 1:

I would encourage for example that the authors recognize more clearly that LFD tests can miss positive tests (unacceptable false negative rates) and refer to this work <https://pmc.ncbi.nlm.nih.gov/articles/PMC8758200/>

- We have added a sentence to the introduction (page 3) clarifying that lower sensitivity of LFD tests will result in false negative test results occurring.*
- The linked reference discusses that the sensitivity of LFD is too low for whole population screening, which is a different context to that of a prevalence study. As such, the reference is not directly applicable as our study uses tests for surveillance, rather than for initiating interventions.*

Ct values, a proxy for viral load, have been shown to have different characteristics by age. For example, evidence points largely to lower viral load in young children (i.e. higher Ct values). This raises the question on how to account for viral load dynamics in the age-stratified results. The authors have briefly mentioned this on Page 21 in the discussion, but I would want to see some sensitivity analysis to lend credibility.

- We agree with the reviewer that viral load trajectories differ across age groups, which results in differences in test sensitivity dynamics across different age groups. The test sensitivity profiles were estimated in a separate paper which we have referenced in the manuscript (Overton et. al.) and were conditional upon the age group of the case to account for age-dependent viral load dynamics. The test sensitivity profiles demonstrate the expected effects of the viral load dynamics, with younger individuals having lower test sensitivity for example.*
- The discussion on page 21 has been amended to make it clearer that the average sensitivity estimated in this manuscript is a result of epidemic dynamics and the age-dependent test sensitivity profiles, and that the age-dependent test sensitivity profiles capture the effects of differing viral load trajectory characteristics across age groups. We have also clarified that the test sensitivity profiles, and duration of positivity estimates are age-stratified throughout the methods, and highlighted the key differences between age groups.*

The limitations associated with participation bias is problematic. This deserves more discussion on whether the incidence, prevalence, and positivity in the older age group is truly representative amongst this group.

- We agree that there are limitations around participation bias and have discussed this further in the manuscript, page 22.*

Limited spatial prevalence is a concern. The authors do take note that the literature shows significant heterogeneity in COVID-19 with respect to spatial factors. I would like to see the incorporation of more spatial elements in the data (e.g. mobility) to account for the shift in dynamics

- Our reference to the literature was intended to show that significant spatial heterogeneity occurred only in the beginning stages of the pandemic, and that limited spatial heterogeneity was observed once mixing had returned to normal levels. We have not been able to find any evidence for substantial spatial heterogeneity in small countries after approximately 2022. Given the highly transmissible nature of SARS-CoV-2, and that the UK is well connected, most theoretical models would not predict substantial spatial heterogeneity of SARS-CoV-2. We have expanded our discussion of the limited spatial differences to make this clearer.*
- Despite this, we agree that inclusion of mobility matrices could be an interesting research avenue to improve the power of the model at smaller spatial scales where spatial heterogeneity could occur. Though, this would be a complex research problem requiring a separate publication; we are not aware*

of any work using spatiotemporal smoothing using mobility matrices and post-stratification. Based on previous experience using mobility matrices for smoothing, we believe that considerable methodological development would need to be undertaken.

With respect to the multilevel regression, the authors pool age, sex, location using an additive logit function and the incidence is restricted to help the chain converge. How was the 5% threshold decided?...

- *We have expanded and clarified our argument for setting the incidence to have an upper limit of 5% in the manuscript.*
- *To summarise here, we consider a 5% incidence rate to be effectively impossible to ever observe in a large population due to epidemic dynamics, such as susceptible depletion, and current levels of immunity. Further, a 5% incidence upper limit implies a prevalence upper limit of around 50% - this is far higher than any observed comparable prevalence estimates for a large population. As such, we believe that this upper limit does not exclude any plausible values of SARS-CoV-2 incidence.*
- *If we allow the model to consider incidence rates above 5%, the model quickly encounters computational issues during initialisation – though there is no clear threshold at which computational issues can begin as the duration of positivity also must be initialised. During experimentation, we found that 5% value was the largest value that allowed the model to initialise efficiently.*
- *The final estimate incidence rates never approached the 5% threshold, peaking at around 0.5%, suggesting that the results are not constrained by the upper limit.*

... Conversely, would MCMC model hold when incidence is very low given additive and independent nature of the inverse logit?

- *The MCMC model would hold if incidence were very low, however the intercept priors would need to be updated accordingly. As the model is on the logistic scale, the additive component is estimating differences in the log-odds between groups (i.e. log of the odds ratio between those groups). The difference in the log-odds ratio between two groups is largely invariant to the scale, therefore an additive logistic model can adequately describe the incidence regardless of whether incidence is small or large.*

Compared to developing a Bayesian multilevel regression with time series convolution to capture repeated LFD testing with all the issues related to selection bias, age-dependent results, and absence of spatial variations, would this effort not be better suited to utilize wastewater-based surveillance data?

- *It is true that complex models must be developed to accurately estimate incidence/prevalence from an LFD-based prevalence survey. However, many of the sources of bias/error/uncertainty were well-known beforehand, and as such our study was designed to allow us to quantify and adjust for many of them.*
- *Wastewater surveillance data also has a lot of sources of bias/error/uncertainty that must be accounted for to produce an accurate prevalence estimate or useful surveillance signal: temperature-dependant viral degradation over transit, heterogenous transit lengths, spatial heterogeneity in pre-treatment, heterogeneity in viral shedding, post-infection shedding, unpredictable extrinsic variables such as rainwater or industry, and substantial uncertainty and temporal variation in the population that contributed to a sample. In our opinion, many of these sources of errors/bias/uncertainty would be difficult to quantify or adjust for.*
- *Overall, we believe that producing reliable surveillance or prevalence estimates from wastewater would be a substantial scientific undertaking, and the resulting study may incur a different set of limitations or uncertainties that could be difficult to address.*

Reviewer 2:

The manuscript includes considerable technical statistical descriptions that may limit readability and accessibility for a broader audience, including policymakers and public health officials. I recommend summarizing key statistical methods and concepts using simpler terms within the methods section, while providing the detailed descriptions in an appendix.

- *We appreciate that this manuscript is quite technical in nature, which may limit its accessibility to a broader audience. While the study was running, UKHSA published the findings every two weeks on the official UK government website. These reports were designed to be easily readable by public health officials, policy makers and members of the public, and contained simpler explanations of the technical details. We have amended the manuscript to clearly highlight these reports and other resources for understanding the findings of the study.*
- *For this manuscript, we intended to provide a rigorous description of the statistical methodology, alongside some additional technical findings from the study that were not previously published. We consider the statistical framework that was developed during this study to be an important scientific output from the study, as it addresses many of the issues that would be encountered if LFD-based prevalence studies were to be performed again in the future. As such, we consider it important to keep the statistical methodology as a core component of the manuscript.*

It is unclear whether prior infection, vaccination history, and recent vaccination were factored into the analysis...

- *It is not possible to produce representative estimates of incidence/prevalence that adjust for prior infection, vaccination history, or recent vaccination. This would require data on the size of each sub-population to be further stratified by these variables, which is not available and so post-stratification could not be used to produce representative estimates that that level of granularity.*

... and to what extent these factors contributed to the observed decline in incidence and prevalence among older individuals, beyond differences in contact rates. Given that older adults may have had higher comorbidity rates and were likely prioritized for vaccination, these factors could have influenced their lower incidence and prevalence. Clarifying how these variables were accounted for would strengthen the epidemiological interpretation of the findings.

- *We agree that higher levels of vaccination in older age groups is likely a contributing factor to the lower incidence/prevalence in those age groups. We have amended our discussion to make this clearer.*
- *The purpose of this study was to produce representative estimates of incidence/prevalence for subpopulations stratified by age/sex/location. Reliable estimates of these quantities are important to public health officials and decision makers. Our estimates of incidence/prevalence are, of course, impacted by vaccination and immunity which differ across age groups, and this is why it was important to have representative age-stratified estimates.*
- *We agree that it would be useful to provide incidence/prevalence stratified by vaccination status or history, though it is unlikely that there is sufficient data to estimate the necessary epidemiological parameters when stratified by vaccination status/history/infection history given the large number of different timings and combinations. It would also require estimating how the vaccine effects wane over time.*

Please specify the extent of non-compliance with repeat testing and whether this varied across age groups, as well as the implications on the study findings.

- *The models used to estimate the epidemiological parameters from the repeat testing data were detailed in a separate manuscript (Overton et. al), which contains further data and details on uptake of repeat testing.*
- *For each age group, there were enough individuals who completed the repeat testing to ensure that estimates of epidemiological parameters are reliable for each age group (discussed in Overton et. al.). While older individuals did submit more repeat tests, the models detailed in Overton et. al. were designed so that this does not bias the estimated epidemiological parameters of other age groups. Censored regression models were used to correct for individuals that did not complete repeat testing. As such, the study findings should not be impacted by differential non-compliance with repeat testing across age groups.*

It is unclear whether repeat testing participation differed between individuals who tested positive in the first round versus those who tested negative, and how any potential bias arising from this was addressed. ...

- *Engagement with repeat testing was lower for those who tested positive in the first round of the study. However, this is not expected to introduce any bias into the study as the epidemiological parameters inferred from the repeat testing data were estimated using repeat testing data from all rounds, and these parameters are not expected to vary over time.*

... If possible, providing a comparison of compliant versus non-compliant individuals in terms of repeat testing, stratified by demographic characteristics, would be valuable. More importantly, assessing differences in vaccination status, comorbidity profile, and prior infection history could help determine whether these factors influenced follow-up testing participation and the overall representativeness of the findings.

- *There were no statistically significant differences in engagement with repeat testing when stratified by: region, sex, ethnicity, IMD decile, comorbidity (long term illness, illness that causes reduced ability, smoking, etc).*
- *Younger individuals were less likely to engage with the repeat testing. However, the epidemiological parameters estimated from the repeat testing data were age-stratified, and for each age group there was a sufficient sample size to reliably estimate the epidemiological parameters. As such, it is unlikely that differential engagement with repeat testing introduced bias, as the models were designed to avoid this source of bias.*
- *At a population level, those with fewer vaccine doses were less likely to engage with repeat testing, however this is likely a result of younger individuals having fewer vaccine doses and being less likely to engage with repeat testing. When stratified by age, the number of vaccine doses did not correlate with engagement with repeat testing.*
- *Unfortunately, we are unable to assess whether prior infection history impacted the uptake of the repeat testing participation as these variables were not collected as part of the study.*
- *The repeat testing models were not estimated in this paper, and such it would be unusual for us to comprehensively discuss the data used in those models in this manuscript. Additional details, such as demographic breakdowns of participants, are available in the manuscript by Overton et. al.*

It would be helpful to clarify whether there were specific criteria for classifying a new positive test in a subsequent wave as a new incident infection. This is particularly relevant in differentiating between reinfections and prolonged viral shedding.

- *When estimating the incidence it is not necessary to distinguish whether a positive test was the result of a new incident infection, or a prolonged episode of viral shedding; either way they are positive, and the model will use that information to infer incidence/prevalence by adjusting for duration of positivity and sensitivity. Therefore, this would probabilistically account for prolonged episodes of viral shedding vs newly incident infections when estimating the incidence.*
- *For the parameter estimation models in Overton et al., if an individual submitted a positive test in consecutive rounds of the study, these were aggregated into a single positive episode.*

Response to reviewers

We would like to thank both reviewers for their time spent reviewing this manuscript, and for their useful comments.

Reviewer 1:

With respect to the first point about the sensitivity of LFD tests, I agree that the reference provided is for whole population screening. Yet, the premise of the current work is at the population level rather than a specific subgroup (e.g. occupational, remote/rural areas). Therefore, I think that positioning the paper as a contribution to the field of population monitoring, screening, surveillance should metrics be LFD-based is important and adding a reference that speaks to the limitation of LFD tests is critical.

- *We agree that it is important to discuss the limitations of LFD tests. During our previous revisions we added sentences highlighting that LFD tests have a lower sensitivity compared to PCR tests, and that this will result in false negatives occurring that need to be corrected for. Additionally, we added a reference to estimates of LFD test sensitivity produced from UK data, as we thought this a more directly applicable reference with a larger dataset and more complete analysis. We believed this would address the reviewers original request to highlight more clearly that LFD tests can have false positives and made the limitations of LFD tests clear.*
- *We would like to avoid conflating our surveillance study with whole population screening by referencing a paper which claims LFD tests have an unacceptable false negative rate for whole population screening. In surveillance, false negative tests require statistical correction only, as no action is taken upon a positive test result. For whole population screening this is not the case, and the impact of false negatives and false positives need to be considered in terms of both the goals of the screening program and outcomes. The idea of an acceptable false negative rate is therefore very different between these two settings.*

The discussions on pages 21 and 22 in relation to age-dependent sensitivity of the testing as well as spatial heterogeneity are very well articulated, thank you.

- *Thank you*

The impact of this paper would be much greater if the code was also available in a GitHub repository.

- *The stan code is being released with the manuscript, alongside a synthetic dataset and a script that fits the model and produces some results.*

Reviewer 2:

The authors have addressed all of my comments. I have no further comments on this manuscript.

- *Thank you.*